bioengineering/biomaterials/cellular biology

microfluidic, single cell, droplets, transfection, hiPSCs

# Single cell transfection of human-induced pluripotent stem cells using a droplet-based microfluidic system

Camilo Pérez-Sosa[1,2], Anahí Sanluis-Verdes[1],
Ariel Waisman[2], Antonella Lombardi[2], Gustavo Rosero[1],
Alejandro La Greca[2], Shekhar Bhansali[4],
Natalia Bourguignon[1,4], Carlos Luzzani[2], Maximiliano.
S. Pérez[3,4], Santiago Miriuka[2] and Betiana Lerner[1,4]

[1]National Technological University (UTN), IREN Center, Buenos Aires, Argentina
[2]National Scientific and Technical Research Council (CONICET) - Foundation for the Fight Against Childhood Neurological Diseases, (LIAN-CONICET-FLENI), FLENI Escobar Headquarters, Route 9 Km 53, 1625, Belén de Escobar, Buenos Aires, Argentina
[3]University of Buenos Aires (UBA), Institute of Biomedical Engineering, Paseo Colon 850, C1428EGA Buenos Aires, Argentina
[4]Department of Electrical and Computer Engineering, Florida International University, Miami, FL 33174, USA

 CP-S, 0000-0002-1953-7671; AL, 0000-0002-1940-0173;
BL, 0000-0003-3341-5686

**Authors for correspondence:**
Santiago Miriuka
e-mail: smiriuka@fleni.org.ar
Betiana Lerner
e-mail: belerner@fiu.edu

Microfluidic tools have recently made possible many advances in biological and biomedical research. Research in fields such as physics, engineering, chemistry and biology have combined to produce innovation in microfluidics which has positively impacted diverse areas such as nucleotide sequencing, functional genomics, single-cell studies, single molecules assays and biomedical diagnostics. Among these areas, regenerative medicine and stem cells have benefited from microfluidics since these tools have had a profound impact on their applications. In this study, we present a high-performance droplet-based system for transfecting individual human-induced pluripotent stem cells. We will demonstrate that this system has great efficiency in single cells and captured droplets, like other microfluidic methods but with lower cost. Moreover, this microfluidic approach can be associated with the PiggyBac transposase-based system to increase its transfection efficiency. Our results provide a starting point for subsequent applications in more complex transfection systems, single-cell differentiation interactions, cell subpopulations and cell therapy, among other potential applications.

# 1. Introduction

The field of regenerative medicine has been radically moved forward with the development of hiPSCs. These cells are obtained by genetically reprogramming terminally differentiated adult cells, and can be later maintained indefinitely in *in vitro* culture and can be differentiated to all the cell types in the adult organism [1]. Thus, these cells provide unprecedented opportunities to study the earliest stages of human development *in vitro*, to model various human diseases, to perform drug tests in culture and, ultimately, as an unlimited source of cells for future therapeutic applications. To take advantage of this potential, it is essential to be able to control the differentiation of hiPSCs to somatic lineages with high efficiency and reproducibility in a scalable and cost effective way [2,3].

Cell-to-cell heterogeneity has been an important discussion factor in the phenotypic–genotypic profile of healthy tissues and tissues with disease/pathology [4,5]. Single cell analysis technologies seek to study this heterogeneity to understand cellular, molecular and tissue physiology. In contrast to bulk tests, heterogeneity measurements through individual cell assays provide a much higher phenotypic resolution and do not require prior knowledge of cell subtypes within a sample [6]. Single cell behaviour depends on the properties of complex niches that provide a wide variety of biochemical and biophysical signals [7]. To unravel and understand the behaviour of these cells, it is necessary to deepen the analysis even to see the unicellular behaviour and the possible cellular subpopulations that exist within apparently homogeneous tissues. However, currently technologies for single cell analysis are highly expensive and their yields are limited, which means that few laboratories have access to this type of analysis.

Droplet microfluidic systems allow the isolation of individual cells in conjunction with reagents in liquid capsules or picolitre monodisperse hydrogels with a yield of thousands of droplets per second [8]. These qualities allow many of the challenges in the analysis of a single cell to be overcome. Monodispersity allows quantitative control of reagent concentrations, while droplet encapsulation provides an isolated compartment for the single cell and its environment [9]. This high performance allows the processing and analysis of the tens of thousands of cells that must be analysed to accurately describe a heterogeneous cell population in order to find uncommon cell types or access sufficient biological space to find successes in a directed evolution experiment [10]. Despite this, there are many challenges that microfluidic systems still have for single cell analysis, among which are the performance of the systems, the cost-benefit, the real-time analysis of each droplet, the storage systems, among others.

Functional genomic studies such as the transfection of fluorescent reporters are essential for unraveling cellular and molecular biological phenomena [11,12]. When trying to generate stable cell lines that incorporate a construct within the genome, the PiggyBac technology based on the system of transposable mobile genetic elements has recently gained ground as an alternative to classical methods such as viral transduction [13]. Briefly, this methodology relies on the transient transfection of a plasmid with the sequence to be inserted flanked by inverted terminal repeat sequences (ITRs) and a transposase enzyme that catalyses the integration of the foreign DNA between AATT chromosomal sequences. This methodology has proven highly efficient in human and other mammalian cell types [13–15]. As a consequence of its effectiveness, this method replicates the DNA inserts in a very heterogeneous way, which in turn suggests that the subsequent functionality may be unspecific for the practical purposes of the technique.

In response to this, numerous techniques have been created in order to increase the efficiency of transfection [16–18]. Individualizing the cells and enclosing them within droplets at the picolitre scale allows the cell to have greater contact with the reagents and plasmids, consequently increasing the transfection efficiency. However, production of hundreds or thousands of droplets can limit the optimal individual follow-up.

In this work, we develop a microfluidic system of two interconnected microdevices that have a high performance at the time of individualizing the cells and a low manufacturing cost compared to similar systems [19–22]. The latter often use SU8 resin as the manufacturing base which is significantly more expensive and a single microdevice to form the droplets and store them; this tends to limit long-term cultivation and future clonal selections. Unlike the methods described above, our method of manufacturing the storage microdevice uses a PDMS multilevel system, which allows the droplets to be stored according to manufacturing size and depth. This method is versatile, allowing the manufacture of droplet storage devices from 30 um to 3 mm of diameter.

The process consists of two stages. First, a microdevice that is responsible for generating monodisperse droplets in which the cells (in our case, hiPSCs) are captured together with the

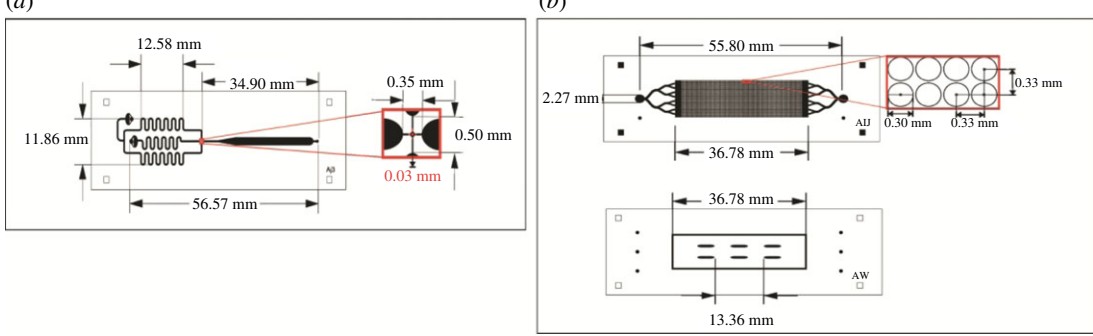

**Figure 1.** (*a*) Shows the diagram of the droplet forming microdevice and the description of the operation of each channel (*b*). It describes the droplet storage microdevice made up of two layers of PDMS, the first layer contains the storage wells, and the second layer is composed of the inlet channels and the support pillars.

plasmids and transfection reagents. The second part consists of a multilevel microdevice designed with thousands of wells where the droplets are stored allowing cell culture and real-time monitoring of cell-to-cell transfection and cell viability over time. In addition, the intrinsic advantage of being separate microdevices allows long-term cultivation and multiple uses of accurate cell visualization tools. We strongly believe that relatively cheap devices could function as a starting point for subsequent applications in more complex transfection systems, unicellular differentiation interactions, or cellular subpopulations, among other potential applications.

## 2. Material and methods

### 2.1. Design and fabrication of microfluidic microdevices

Two separate microdevices were designed for droplet formation and storage applications (figure 1). The droplet forming microdevice was manufactured using conventional, glass-based technology. The droplet storage microdevice was made up of two layers of PDMS in which the first layer had the channels through which the droplets go and the second the wells where they are stored. This device was manufactured with multilevel technology. The dimensions of the two microdevices were $76 \times 26$ mm and their height was 3 mm. The microdevices were manufactured as described previously [23], in the case of the forming microdevice of droplets. Briefly, a mould in high relief with the desired design was made by photolithography in a 700 μm thick silicon wafer (Virginia Semiconductor, Inc.), using the negative resin SU-8 (MicroChem). The microchannels have a final height of 150 μm. Next, the mould was placed under vacuum with trichloro (1H, 1H, 2H, 2H-perfluoro-octyl) silane (Sigma) for 1 h, to protect the SU-8 resin from detachment by releasing PDMS from the mould. The PDMS was mixed with the curing agent in a 10 : 1 ratio, and the mixture was placed under vacuum for 1 h to remove air bubbles. Next, the mixture was poured back under vacuum for 1 h, and cured in an oven at 70°C for 70 min. The PDMS was not moulded, and the fluidic connection ports were constructed by drilling holes in the PDMS with a syringe needle (21-gauge, internal diameter of 0.51 mm). Finally, the PDMS was assembled with the glass base. Through the plasma oxygen system (deposition of chemical vapours enhanced with plasma), the microdevice and the glass base are exposed for 3 min at a pressure of more than 4000 g overnight.

In the case of the droplet storage microdevice, a new custom made multilevel microfluidic manufacturing technology was used [24]. Briefly, the droplet storage microdevice was designed with a Layout Editor design editor software and transferred to the TIL with a 2400 ppi infrared laser source, then the TIL was laminated on the unexposed photopolymer plate, the photopolymer plate was exposed at UVA light at 0.45 J on the back for 10 s, a part of the photopolymer was covered with a mask plate on the back side, the photopolymer plate was exposed to UVA light at 0.45 J on the back side for 20 s, the previous exposures were repeated one at a time, and finally the front side was exposed to UVA light at 19 J for 360 s. After the TIL was removed, the plate was washed with PROSOL N-1 solvent (supplied by Eastman Kodak) at 360 mm min$^{-1}$ and was dried in an oven for 30 min at 50°C, the plate had a last exposure to UVC light at 10 J for 17 min and UVA light was applied at 4 J for 2 min on the front side. With the mould manufactured briefly, a mixture of epoxy

resin and curing agent (Cristal-Tack, Novarchem, Argentina) was poured over the female mould to replicate the design in high relief.

After curing, the epoxy resin mould (ERmold) was detached from the Fmold to form the male mould. Subsequently, a mixture of PDMS and curing agent in a 10 : 1 weight ratio (Sylgard 184 silicone elastomer kit, Dow Corning) was poured onto the ER mould and cured in an oven at 40°C overnight. The same steps described for the droplet forming microdevice were used for assembly.

## 2.2. Production of droplets storage and breaking

For the production of monodisperse droplets, the biocompatible oil FluoroSurfactant-HFE 7500 5% (Ran Biotechnologies, Beverley, MA) was used for the continuous phase, and for the dispersed phase the cells were used in suspension in mTeSR™1 (STEMCELL Technologies) medium with Rock Inhibitor (Y-27632). The injection of the phases was performed using the infusion set Fullgen A22 (USA), with results for the continuous phase of $4.50\,\mu l\,min^{-1}$–$6.50\,\mu l\,min^{-1}$, and for the dispersed phase $1.75\,\mu l\,min^{-1}$–$3.00\,\mu l\,min^{-1}$. The average size of the droplets generated was $80\,\mu m$–$87\,\mu m$ (s.d.: 0,4358), then a tube was connected to the output hole of the producer microdevice and the inlet hole of the droplets storage microdevice was progressively increased to complete the storage volume.

In order to release the cells that were encapsulated within the oil droplets, SIGMA 1H, 1H, 2H, 2H perfluoro-octanol (PFO) was used based on the protocols recommended [25]. In summary, the droplets that were in the micro storage device were collected. After this, base medium (E8-Flex) and a volume like that collected in the droplets of (PFO) was added. Using light movements, it was mixed and left to rest at room temperature for 2 min. After seeing the two separate phases (oil/medium), it was centrifuged with a short spin for 10 s. The supernatant containing the cells was taken for analysis or common plating.

## 2.3. Droplet images

A Nikon SMZ645 Stereo Zoom Microscope and a Canon T3-I Rebel digital camera connected to the microscope were used to capture the droplets. The images were created from a stack of multiple acquisitions of microscopes in a large surface area of the microdevice.

The analysis and quantification of the results obtained in the experimental phases were performed using the ImageJ software and R.

## 2.4. Two-dimensional cell culture

The hiPSCs cell line used in this study was generated in our laboratory from adult male fibroblasts and was previously described [26]. Cells were regularly cultured in six well plates in E8-Flex medium with Geltrex coating (all Thermo Fischer Scientific). Cells were passaged every 2–3 days using Tryple until the moment of encapsulation and subsequent transfection. All cell cultures were maintained at 37°C in a saturated atmosphere of 95% air and 5% $CO_2$.

## 2.5. Plasmids used in this study

To assess the transfection efficiency and to generate stable cell lines, two plasmids were used [27]. A first plasmid that encodes a CAG-mCerulean-H2B reporter together with a Neomycin resistance cassette, both flanked by ITR sequences. The second plasmid drives the transient expression of the PiggyBac transposase.

## 2.6. Cell viability and nuclear staining

To measure cell viability after droplets encapsulation and transfection, we used the Live Dead Assay kit (Thermofisher Scientific). Since the droplets-based system differs according to standard culture techniques, we empirically determined that five times more reagent was required than recommended by the manufacturer. Hoechst 33 342 was used for staining cell nuclei in living cells to measure transfection efficiency.

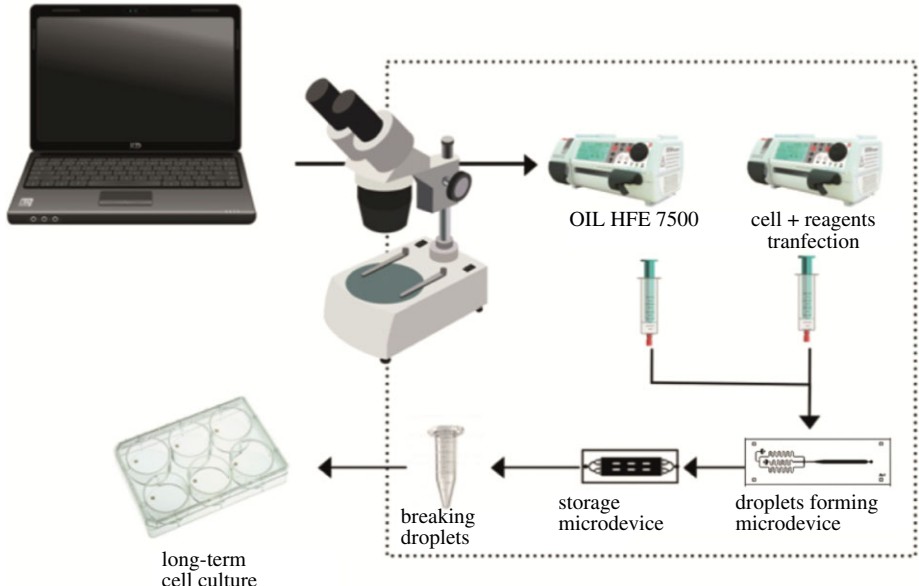

**Figure 2.** Scheme of the microfluidic transfection system.

## 2.7. Trypan blue staining

To measure the viability of the culture as a long-term stable line, cells were harvested and stained with a 0.4% trypan blue solution (0.08% final concentration) (Sigma, St. Louis, MO, USA) for 5 min at room temperature. Cells-stained blue dead and unstained live were counted in a hemocytometer chamber. The percentages of viable cells were calculated as the total number of living cells divided by the total number of cells and multiplied by 100.

# 3. Results and discussion

## 3.1. Microfluidic system design

The constructed single cell microfluidic transfection system is illustrated in figure 2. This system consists of two interconnected microdevices and an outlet channel. The first produces monodisperse droplets containing cells, plasmids and transfection agents using the flow-focusing system. These flows are controlled by infusion pumps with infusion and pressure control. The second microdevice is made up of two layers of PDMS that have 3000 multi-level wells that allow a high droplet capture performance and an output channel to recover them. In addition, this double-layer design allows for greater gas exchange in conjunction with fluorinated oil (HFE-7500), which greatly helps cell viability [28].

## 3.2. Droplet production, storage and encapsulation

First, the efficiency of the microdevices that make up our system was characterized. Figure 3*a* shows the production of droplets in the outlet channel of the forming microdevice. There were a total of 1000 droplets of average size of 83 µm with an s.d. of 0.4358 (figure 3*b*). This demonstrated that the device is effective to generate monodisperse droplets, and that most of the droplets had a similar size. The perfusion rate was 6.50 ul min$^{-1}$ for the continuous phase (oil) and 3.00 ul min$^{-1}$ for the dispersed phase (cells, plasmids and transfection agents). Furthermore, together with these variables, cell concentration solutions were enhanced by applying the Poisson probability distribution, with an optimal single cell encapsulation result of 300 000 cells ml$^{-1}$.

On the other hand, the efficacy of the microdevice that stores the droplets was measured (figure 5*a*). High efficiency is determined by the storage microdevice to capture one droplet per well. This is remarkably helpful during image analysis and real-time cell detection. In figure 4*b*, the wells of the storage microdevice were observed capturing the droplets with cells inside.

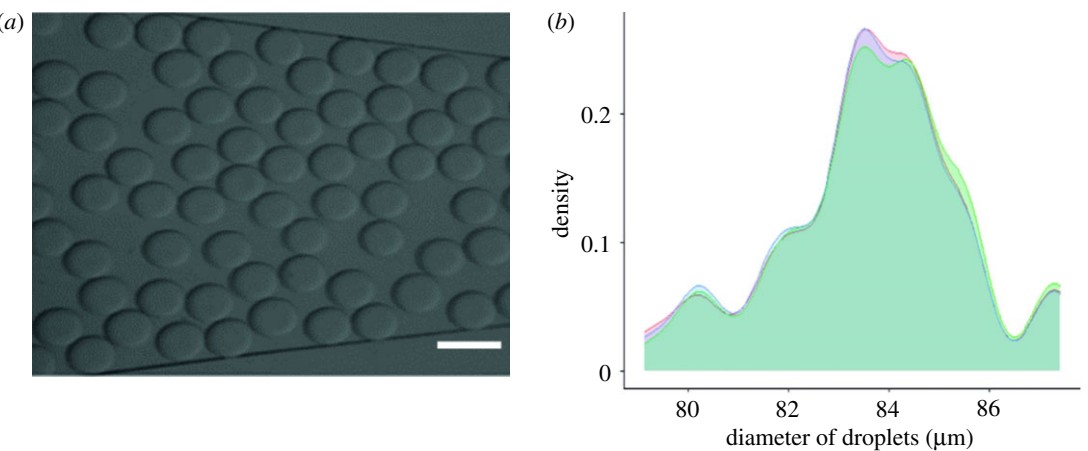

**Figure 3.** Droplet forming. (*a*) Representative image of the droplets produced in the outlet channel of the forming microdevice. (*b*) The density graph shows the size of the droplets produced by the system. Each line represents each technical sample that was made. The scale bars correspond to 100 µm.

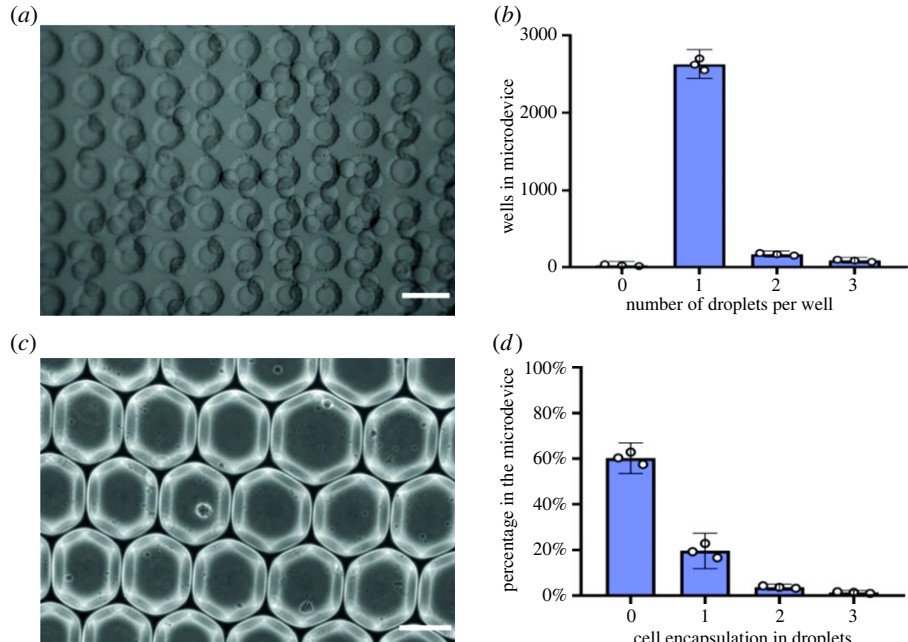

**Figure 4.** Characterization of the microfluidic system. (*a*) Representative image of droplets captured in the micro storage device. (*b*) Number of droplets captured per well of the micro storage device. White points indicate the mean value for each replicate (*N* = 3). (*c*) Representative image of cells inside that travel through the outlet channel of the forming device. (*d*) The bar graph shows the effectiveness of capturing a cell per droplet. White points indicate the mean value for each replicate (*N* = 3). The scale bars correspond to 100 µm.

We next evaluated the level of encapsulation of cells per droplet. Figure 4*c* shows cells encapsulated within monodisperse droplets formed in the system. As expected, most of the droplets did not contain any cells, while we observed an efficiency of 22% in single cell encapsulation (figure 4*d*). These values resemble similar work of encapsulation and storage of droplets [29]. However, the system that we present is ahead of other similar systems due to the versatility of its design that can store droplets of various sizes and volumes with fast and low-cost manufacturing.

## 3.3. Cell viability in the microfluidic system

It is widely known that hiPSCs cells are highly sensitive to changes in their environment that can reduce their viability and even prompt them to enter pathways of apoptosis [30]. First, to demonstrate the effectiveness of our system, the viability of cells was evaluated at different times after encapsulation

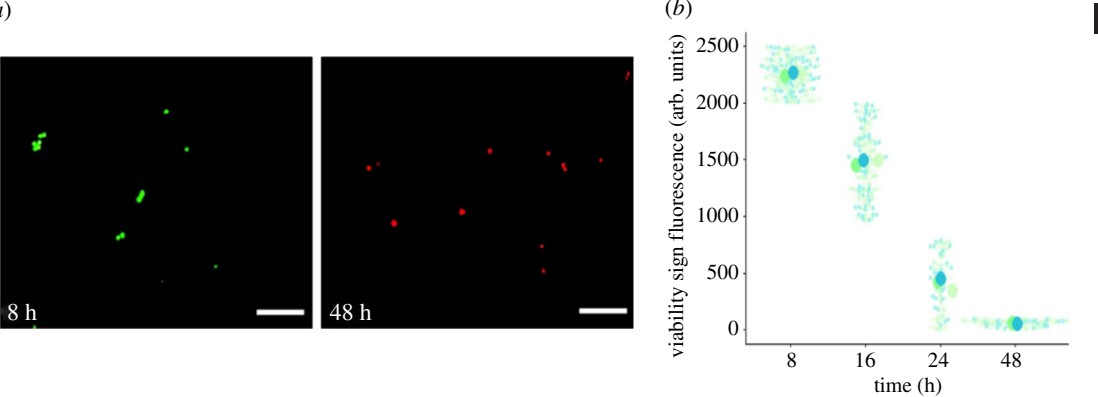

**Figure 5.** Cell viability. (*a*) Representative image of cell viability. Green fluorescence indicates cells with marked cell viability, while red fluorescence shows cells that are not viable or in an apoptotic state (*b*). Cell viability as a function of encapsulation time. Small points correspond to mean intensity measurement of individual cells within each biological replicate as indicated by different colours (*N* = 3). Larger points depict mean value for all measurements in each specific biological replicate. The scale bars correspond to 200 µm.

and subsequent release of droplets. For this purpose, we employed the methodology of the Live Dead test kit, which emits different fluorescent signals depending on the state of viability of the cells (figure 5*a*). We observed high cell viability at 8 and 16 h. However, it became clear that cell viability was inversely related to the time spent inside the droplet. At 24 h of encapsulation and at 48 h, a high degree of cell death and low viability with time was evident (figure 5*b*).

## 3.4. Efficient transfection of hiPSCs using the microfluidic system

Cell transfection is an invaluable tool as it allows the generation of engineered cell lines with Crispr/Cas9, the overexpression of proteins, gene silencing, fluorescent reporter visualization, among many others. Due to hiPSCs displaying high cell viability during the first 8 h after encapsulation, we decided to evaluate the transfection efficiency of the cells within the droplets. Therefore, transfection of a fluorescent H2B-mCerulean indicator was evaluated when the transfection reagents were provided during the encapsulation step.

After cell encapsulation and droplet storage, the microdevice was placed under culture conditions at 37°C in an atmosphere saturated with 95% air and 5% $CO_2$. To assess transfection efficiency, the amount of fluorescence emitted by each individual cell within the droplet was measured in a period of 2, 4, 6, 8 h (figure 6*a,b*). Fluorescence intensity increased with time after encapsulation, with higher levels at 6 and 8 h. This observation could be explained both because of the time it takes for fluorescent proteins to be transcribed, translated, and matured, as well as by the increased likelihood of cell transfection when cells are confined within the droplets with the transfection reagents. Indeed, our microfluidic system is suitable for single cell transfection and it could be potentially applied in different methodologies.

## 3.5. Comparison of plate methods with the microfluidic system

To further characterize our microfluidic transfection set-up, the system was compared with conventional plate transfection method in multiwell plates. As shown in the previous transfection experiment, the optimal encapsulation time considering cell viability and transfection was 8 h. In this context, we compared both methodologies with the same transfection time and with the same initial concentrations of plasmids and lipofectamine.

Figure 6*c* shows the comparison of both methods. As there is a significant proportion of transfected cells in both strategies, results indicate a significantly higher intensity level of fluorescence in the microfluidic system (figure 6*c,d*), which could be a consequence of the confinement of the transfection reagents within the droplets.

Although cell viability was reduced after prolonged culture within the droplets, our results show that transfection of hiPSCs in our microfluidic system was comparable and even displayed higher levels of transfection compared to the conventional methodology in multiwell plates. In figure 6*e*, we show the transfection efficiency of each method in percentage. An efficiency in the plate method of around 60%

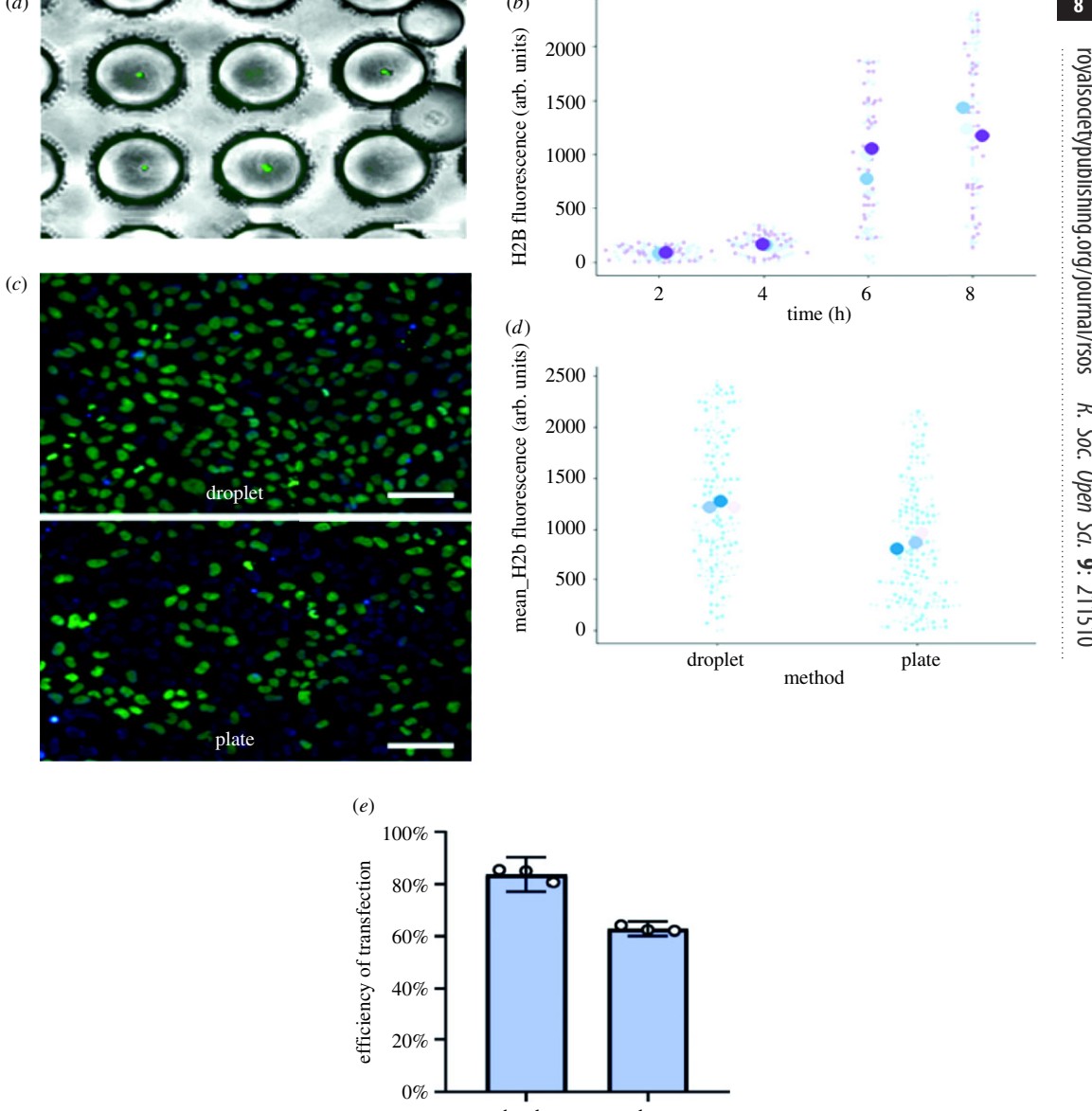

**Figure 6.** Transfection and method efficiency. (*a*) Representative image of cells within the droplets. Each drop is captured in a well of the storage microdevice. (*b*) H2B-mCerulean fluorescence at different times after encapsulation with transfection reagents. (*c*) The images show the comparative top platelet of the cells transfected into droplets and subsequently released. The image below corresponds to the traditional culture plate transfection method. (*d*) Comparison of H2B-mCerulean transfection in the microfluidic system or in multiwell plates. In the microfluidic system, cells were encapsulated with transfection reagents and the reporter plasmid for 8 h, droplets were broken, and cells were later cultured in multiwell plates until fluorescence was measured 24 h after encapsulation. In the case of multiwell transfection, reagents were incubated during 8 h, medium was changed, and fluorescence was evaluated at 24 h. (*e*) Comparison of transfection efficiency between the two methods. The scale bars correspond to 200 µm.

was observed; by contrast, the efficiency of the droplet method exceeded 80%. All experiments had technical replicates ($N = 3$), represented by the white circles in figure 6*e*.

## 3.6. Post-transfection cell viability

To potentially generate a stable cell line, the viability of the cells was measured. For this, after the transfection and release of the cells that were in the droplets, these were seeded in a conventional cell culture plate. Survival was analysed using the trypan blue (exclusion dye) counting method, during three reseeds. Cells remained viable during the three reseeds as shown in table 1 and viability

**Table 1.** Shows the viability of the cells during three passages, post-transfection and release of the cells through the droplet system.

| transfected cells | cells counted | percentage of viability (%) |
|---|---|---|
| cell passage 1 | 134 | 87 |
| cell passage 2 | 145 | 92 |
| cell passage 3 | 166 | 95 |

improved with each cell passage. This demonstrated that the system works for further analysis or stable line constructions. In the electronic supplementary material, the images show the cell count in each passage.

## 4. Conclusion

The developed droplet-based microfluidic system showed high transfection efficiency of hiPSCs cells, by inserting the H2b-cerulean fluorescent protein. We hypothesized that this could be due to the microenvironment conditions during encapsulation, in which cells were exposed to plasmids and transfection reagents for long periods of time. The microfluidic system also proved to be highly effective in capturing one droplet per well and individualizing cells within the droplets for subsequent real-time monitoring. This is attributed to the multi-level storage microdevice, which allows the operator to visualize and determine the location of each cell during the transfection process.

Single cell transfection performed by the microfluidic droplet method was shown to be highly effective and economical compared to other similar methods. We also highlight the versatility of the storage device that can store droplets of different sizes, with different systems of entry channels and with different objectives not only focused on biological areas.

We believe that our microfluidic system could be a powerful tool for assays where isolation and transfection of individual cells are needed, such as the generation of clonal cell lines using CRISPR/Cas9 technology, regenerative medicine and gene therapy by providing homogeneity because of transfecting individual cells. While our work focused on genome editing by transfection, this innovative system can be used for various purposes such as the production of proteins from individual cells, microbioreactors, selective isolates, drug testing, among other applications at costs that most laboratories could afford.

Data accessibility. The data from the droplet measurements, the characterization of the store microdevice, and the comparison of transfection methods, along with the IMAGE J and r analysis scripts are available at the following link: https://doi.org/10.6084/m9.figshare.13008041.v1.

Authors' contributions. All authors gave final approval for publication and agreed to be held accountable for the work performed therein.

Competing interests. We declare we have no competing interests.

Funding. The authors received financial support from CONICET, ANPCyT and Biothera Foundation.

Acknowledgements. The authors are thankful for the financial support from CONICET, ANPCyT and Biothera Foundation. We thank Jorge. L. Fernandez and I. de Sá Carneiro for general support and fruitful discussions. We also want to thank the UTN-FRGP design and communication team, for their valuable collaboration.

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
