## [Peer Review File · Royal Society Open Science]

Review History

RSOS-201781.R0 (Original submission)

Review form: Reviewer 1

Is the manuscript scientifically sound in its present form?

Yes

Are the interpretations and conclusions justified by the results?

Yes

Is the language acceptable?

Yes

Do you have any ethical concerns with this paper?

No

Have you any concerns about statistical analyses in this paper?

No

Recommendation?

Accept with minor revision (please list in comments)

Comments to the Author(s)

More information needed:

- Some wording seems odd or misplaced
- Lacking of reference papers
- Better to say droplet than drop
- Schematics of the microfluidic chips used (with dimensions and design)
- Concentration of cell solution in the "Cell culture" part in Materials and Methods (poisson law distribution used?)
- Probably more information on the Materials and Methods part (for example, what is the supplier of the surfactant?)
- Figures with more details + axes graphs illisible
- Figure 1: Schematics or pictures of the setup with the microfluidic chips (both of them)?
- Figure 2: mean diameter and standard deviation of droplets size? Schematic of the droplet generation chip?
- Figure 3: possible to have the poucentage of encapsulation for each value: 0 cell, 1 cell, 2 cells, 3 cells?
- Figure 4: possible to have a bright field image to better understand the fluorescence images?
- Figure 5: Are the droplets broken to take the C part? If yes, what is the protocol?

Review form: Reviewer 2

Is the manuscript scientifically sound in its present form?

No

Are the interpretations and conclusions justified by the results?

No

Is the language acceptable?

No

Do you have any ethical concerns with this paper?

No

Have you any concerns about statistical analyses in this paper?

No

Recommendation?

Reject

Comments to the Author(s)

The manuscript describes a microfluidic approach for single cell transfection. A flow-focusing droplet generator and a droplet-storage device were interconnected for the encapsulation of hiPSCs in droplets and storing the droplets in microwells. The authors evaluated the viability of cells in droplets and compared the efficiency of the transfection of hiPSCs between the proposed microfluidic-based approach and the conventional method. However, the devices in the manuscript are well-known designs through many previous studies, and even a chip combined droplet generation and storage is commercially available (Fluidic 488, microfluidic ChipShop,

Germany) for single cell analysis. Also, the authors proposed that the microfluidic method was highly effective for single cell transfection; however, the cell viability in droplets is extremely low for long-term cell culture and the cell viability after transfection was not proven. Hence, unfortunately, this reviewer feels that the manuscript fails to make a strong case that would be suitable for publication in Royal Society Open Science.

- 1) (page 4, line 31) "Biocompatible oil FluoroSurfactant-HFE 7500 5%" is the name of a product, HFE 7500 oil with 5% fluorinated surfactants. Please clarify it.
- 2) (page4, line 35) Please provide the average size of droplets with standard deviation.
- 3) (page4, line 44) The microscope equipped with the camera might be used to capture images of droplets.
- 4) (page5, line 4) "4.1 Microfluidic system design" needs to rearrange in Materials and Methods. If the authors provide CAD designs of the microfluidic devices, it will be informative for readers.
- 5) (page 5, line14) The authors mentioned, "the efficiency of the device was characterized. It would be the capability of the device for generating monodispersed droplets.
- 6) (page 5, line 27) It was reported that the rate of single cell encapsulation in droplets is higher than 22% previously (Collins, D.J., Neild, A., DeMello, A., Liu, A.Q. and Ai, Y., 2015. The Poisson distribution and beyond: methods for microfluidic droplet production and single cell encapsulation. Lab on a Chip, 15(17), pp.3439-3459). Please clarify it.
- 7) (page5, line 38) After 24 hours of encapsulation, a high degree of cell death was observed; however, the authors proposed the microfluidic platform for long-term cell culture. Please explain further how the platform can be used for single cell long-term culture and transfection applications.
- 8) (page5, line 55) The authors showed the efficiency of transfection in the proposed platform. However, cell viability after transfection is one of the most important parameters for single cell transfection applications. Could the authors evaluate the cell viability after transfection in the proposed system?
- 9) (Figure 4) It is not clear where droplets and cells are in figure 4A. A bright-field microscopy image on the area is necessary to be provided. Also, are the two images (green and red fluorescence) combined images? In Figure 4B, the authors showed the fluorescence intensity degradation to quantify the cell viability. However, the fluorescence intensity can be decreased by photobleaching. Hence, if the authors would evaluate the cell viability as a function of encapsulation time, the y-axis of the graph should be the number of cells (fluorescence dots).
- 10) English should be updated. The manuscript is hard to read.

Decision letter (RSOS-201781.R0)

Dear Dr Lerner

The Editors assigned to your paper RSOS-201781 "Single cell transfection of human induced pluripotent stem cells using a droplet-based microfluidic system." have made a decision based on their reading of the paper and any comments received from reviewers.

Regrettably, in view of the reports received, the manuscript has been rejected in its current form. However, a new manuscript may be submitted which takes into consideration these comments.

We invite you to respond to the comments supplied below and prepare a resubmission of your manuscript. Below the referees' and Editors' comments (where applicable) we provide additional requirements. We provide guidance below to help you prepare your revision.

Please note that resubmitting your manuscript does not guarantee eventual acceptance, and we do not generally allow multiple rounds of revision and resubmission, so we urge you to make every effort to fully address all of the comments at this stage. If deemed necessary by the Editors, your manuscript will be sent back to one or more of the original reviewers for assessment. If the original reviewers are not available, we may invite new reviewers.

Please resubmit your revised manuscript and required files (see below) no later than 10-Oct-2021. Note: the ScholarOne system will 'lock' if resubmission is attempted on or after this deadline. If you do not think you will be able to meet this deadline, please contact the editorial office immediately.

Please note article processing charges apply to papers accepted for publication in Royal Society Open Science (<https://royalsocietypublishing.org/rsos/charges>). Charges will also apply to papers transferred to the journal from other Royal Society Publishing journals, as well as papers submitted as part of our collaboration with the Royal Society of Chemistry (<https://royalsocietypublishing.org/rsos/chemistry>). Fee waivers are available but must be requested when you submit your manuscript (<https://royalsocietypublishing.org/rsos/waivers>).

Thank you for submitting your manuscript to Royal Society Open Science and we look forward to receiving your resubmission. If you have any questions at all, please do not hesitate to get in touch.

on behalf of Dr James Locke (Associate Editor) and Pietro Cicuta (Subject Editor)
openscience@royalsociety.org

Associate Editor Comments to Author (Dr James Locke):

Comments to the Author:

Although both reviewers appreciated the work, there were concerns on the novelty of the approach when compared to other published and available techniques. These concerns preclude publication at this time, although a resubmitted manuscript that addressed the reviewers' concerns would be considered.

Reviewer comments to Author:

Reviewer: 1

Comments to the Author(s)

More information needed:

- Some wording seems odd or misplaced
- Lacking of reference papers
- Better to say droplet than drop
- Schematics of the microfluidic chips used (with dimensions and design)
- Concentration of cell solution in the "Cell culture" part in Materials and Methods (Poisson law distribution used?)
- Probably more information on the Materials and Methods part (for example, what is the supplier of the surfactant?)
- Figures with more details + axes graphs illegible
- Figure 1: Schematics or pictures of the setup with the microfluidic chips (both of them)?
- Figure 2: mean diameter and standard deviation of droplets size? Schematic of the droplet generation chip?
- Figure 3: possible to have the percentage of encapsulation for each value: 0 cell, 1 cell, 2 cells, 3 cells?
- Figure 4: possible to have a bright field image to better understand the fluorescence images?
- Figure 5: Are the droplets broken to take the C part? If yes, what is the protocol?

Reviewer: 2

Comments to the Author(s)

The manuscript describes a microfluidic approach for single cell transfection. A flow-focusing droplet generator and a droplet-storage device were interconnected for the encapsulation of hiPSCs in droplets and storing the droplets in microwells. The authors evaluated the viability of cells in droplets and compared the efficiency of the transfection of hiPSCs between the proposed microfluidic-based approach and the conventional method. However, the devices in the manuscript are well-known designs through many previous studies, and even a chip combined droplet generation and storage is commercially available (Fluidic 488, microfluidic ChipShop, Germany) for single cell analysis. Also, the authors proposed that the microfluidic method was highly effective for single cell transfection; however, the cell viability in droplets is extremely low for long-term cell culture and the cell viability after transfection was not proven. Hence, unfortunately, this reviewer feels that the manuscript fails to make a strong case that would be suitable for publication in Royal Society Open Science.

- 1) (page 4, line 31) "Biocompatible oil FluoroSurfactant-HFE 7500 5%" is the name of a product, HFE 7500 oil with 5% fluorinated surfactants. Please clarify it.
- 2) (page 4, line 35) Please provide the average size of droplets with standard deviation.
- 3) (page 4, line 44) The microscope equipped with the camera might be used to capture images of droplets.
- 4) (page 5, line 4) "4.1 Microfluidic system design" needs to be rearranged in Materials and Methods. If the authors provide CAD designs of the microfluidic devices, it will be informative for readers.
- 5) (page 5, line 14) The authors mentioned, "the efficiency of the device was characterized. It would be the capability of the device for generating monodispersed droplets."
- 6) (page 5, line 27) It was reported that the rate of single cell encapsulation in droplets is higher than 22% previously (Collins, D.J., Neild, A., DeMello, A., Liu, A.Q. and Ai, Y., 2015. The Poisson distribution and beyond: methods for microfluidic droplet production and single cell encapsulation. *Lab on a Chip*, 15(17), pp.3439-3459). Please clarify it.

7) (page5, line 38) After 24 hours of encapsulation, a high degree of cell death was observed; however, the authors proposed the microfluidic platform for long-term cell culture. Please explain further how the platform can be used for single cell long-term culture and transfection applications.

8) (page5, line 55) The authors showed the efficiency of transfection in the proposed platform. However, cell viability after transfection is one of the most important parameters for single cell transfection applications. Could the authors evaluate the cell viability after transfection in the proposed system?

9) (Figure 4) It is not clear where droplets and cells are in figure 4A. A bright-field microscopy image on the area is necessary to be provided. Also, are the two images (green and red fluorescence) combined images? In Figure 4B, the authors showed the fluorescence intensity degradation to quantify the cell viability. However, the fluorescence intensity can be decreased by photobleaching. Hence, if the authors would evaluate the cell viability as a function of encapsulation time, the y-axis of the graph should be the number of cells (fluorescence dots).

10) English should be updated. The manuscript is hard to read.

===PREPARING YOUR MANUSCRIPT===

===PREPARING YOUR REVISION IN SCHOLARONE===

Author's Response to Decision Letter for (RSOS-201781.R0)

See Appendix A.

RSOS-211510.R0

Review form: Reviewer 1

Is the manuscript scientifically sound in its present form?

Yes

Are the interpretations and conclusions justified by the results?

Yes

Is the language acceptable?

Yes

Do you have any ethical concerns with this paper?

No

Have you any concerns about statistical analyses in this paper?

No

Recommendation?

Accept as is

Comments to the Author(s)

Nothing more to add

Review form: Reviewer 3

Is the manuscript scientifically sound in its present form?

Yes

Are the interpretations and conclusions justified by the results?

Yes

Is the language acceptable?

Yes

Do you have any ethical concerns with this paper?

No

Have you any concerns about statistical analyses in this paper?

No

Recommendation?

Accept as is

Comments to the Author(s)

The revised version looks good. Agree to accept.s

Decision letter (RSOS-211510.R0)

Dear Dr Lerner,

I am pleased to inform you that your manuscript entitled "Single cell transfection of human induced pluripotent stem cells using a droplet-based microfluidic system." is now accepted for publication in Royal Society Open Science.

on behalf of Dr James Locke (Associate Editor) and Pietro Cicuta (Subject Editor)
openscience@royalsociety.org

Associate Editor Comments to Author (Dr James Locke):

The revised manuscript has been approved by the two reviewers and is now suitable for publication.

Reviewer comments to Author:

Reviewer: 1

Comments to the Author(s)

Nothing more to add

Reviewer: 3

Comments to the Author(s)

The revised version looks good. Agree to accept.

Appendix A

Reviewer comments to Author:

Reviewer: #1

Comments to the Author(s)

More information needed:

1) Some wording seems odd or misplaced

Answer:

English and grammar was checked and corrected in the manuscript.

2) Lacking of reference papers

Answer:

New reference was included in the manuscript, emphasizing the introduction, and focusing on the most current droplet storage methods and transfection techniques. Other bibliography and cited paragraphs were excluded from the text to improve reading comprehension and focus on the advantages of our method compared to those currently available.

New references:

(Headen, García and García, 2018)

(Sun et al., 2014)

(Tsukiyama et al., 2011)

(Ishida et al., 2018)

(Stevenson et al., 2010)

(Bire et al., 2013)

(Woltjen et al., 2011)

3) Better to say droplet than drop

Answer:

This suggestion was taken and corrected in the manuscript.

4) Schematics of the microfluidic chips used (with dimensions and design)

Answer:

The diagram together with the measurements of both microdevices is described in Figure 1.

Figure 1. A Shows the diagram of the droplet forming microdevice and the description of the operation of each channel B. It describes the droplet storage microdevice made up of two layers of PDMS, the first layer contains the storage wells, and the second layer is composed of the inlet channels and the support pillars.

5) Concentration of cell solution in the "Cell culture" part in Materials and Methods (poisson law distribution used?)

Answer:

To find the optimal cell concentration of our droplet forming microdevice, we applied the Poisson distribution was applied based on a cell concentration curve giving the optimal concentration for single cell encapsulation per droplet. This was also explained in the manuscript, giving the optimal concentration of 300,000 cells / ml.

6) Probably more information on the Materials and Methods part (for example, what is the supplier of the surfactant?)

Answer:

Surfactant supplier added in manuscript. "For the production of monodisperse droplets, the biocompatible oil FluoroSurfactant-HFE 7500 5% (RAM BIOTECNOLOGHIES)"

7) Figures with more details + axes graphs illegible

Answer:

The size and resolution of the figures were modified so that the graphs and axes can be read more clearly.

8)Figure 1: Schematics or pictures of the setup with the microfluidic chips (both of them)?

Answer:

It was added in the segment of (figure 1) in materials and methods. All the details of the micro droplet formation and storage devices.

9) Figure 2: mean diameter and standard deviation of droplets size? Schematic of the droplet generation chip?

Answer:

The standard deviation (SD) of the droplets was also detailed in the materials and methods and in the results. Demonstrating the monodispersity of the droplets at the time of their formation in the microdevice.

10) Figure 3: possible to have the poucentage of encapsulation for each value: 0 cell, 1 cell, 2 cells, 3 cells?

Answer:

It is possible to have the total percentage of the wells that had cells, but it was decided to put the total of droplets that were captured in the experimental replicas. So, it was considered more appropriate to present it that way. The point raised by the Reviewer is substantially valid though, that is why we have included it as part of the discussion

11) Figure 4: possible to have a bright field image to better understand the fluorecence images?

Answer:

The Live-Death protocol is designed to view images of cells in fluorecence. The manuscript details that the cells were previously released from the droplets by (PFO) to measure their viability.

12) Figure 5: Are the droplets broken to take the C part? If yes, what is the protocol?

Answer:

It is detailed in the materials and methods segment. The paragraph is detailed below:

In brief, the droplets that were in the micro storage device were collected. After this, base medium (E8-Flex) and a volume like that collected in the droplets of (PFO) was added. Using light movements, it was mixed and left to rest at room temperature for 2 minutes. After seeing the two separate phases (oil / medium), it was centrifuge with a short spin for 10 seconds. The supernatant containing the cells was taken for analysis or common plating.

Reviewer: #2

Comments to the Author(s)

The manuscript describes a microfluidic approach for single cell transfection. A flow-focusing droplet generator and a droplet-storage device were interconnected for the encapsulation of hiPSCs in droplets and storing the droplets in microwells. The authors evaluated the viability of cells in droplets and compared the efficiency of the transfection of hiPSCs between the proposed microfluidic-based approach and the conventional method. However, the devices in the manuscript are well-known designs through many previous studies, and even a chip combined droplet generation and storage is commercially available (Fluidic 488, microfluidic ChipShop, Germany) for single cell analysis. Also, the authors proposed that the microfluidic method was highly effective for single cell transfection; however, the cell viability in droplets is extremely low for long-term cell culture and the cell viability after transfection was not proven. Hence, unfortunately, this reviewer feels that the manuscript fails to make a strong case that would be suitable for publication in Royal Society Open Science.

1) (page 4, line 31) “Biocompatible oil FluoroSurfactant-HFE 7500 5%” is the name of a product, HFE 7500 oil with 5% fluorinated surfactants. Please clarify it.

Answer:

Following the reviewer's recommendation, the supplier and more details of the oil used in the droplet formation experiments were added.

"For the production of monodisperse droplets, the biocompatible oil FluoroSurfactant-HFE 7500 5% (RAM BIOTECNOLOGHIES)"

2) (page4, line 35) Please provide the average size of droplets with standard deviation.

Answer:

The SD of the droplets was detailed in the materials and methods as well in the results. Demonstrating the monodispersity of the droplets at the time of their formation in the microdevice.

"A total of 1000 droplets of average size of 83 μm with an SD of 0.4358"

3) (page4, line 44) The microscope equipped with the camera might be used to capture images of droplets.

Answer:

Yes, the system has a microscope and an adapted camera where the formation of the droplets can be observed. In this case, it was used to monitor droplet formation and subsequent storage.

4) (page5, line 4) “4.1 Microfluidic system design” needs to rearrange in Materials and Methods. If the authors provide CAD designs of the microfluidic devices, it will be informative for readers.

Answer:

Following the recommendation of the reviewer, the details of measurement and design of the microdevices were added to the manuscript in figure 1.

5) (page 5, line14) The authors mentioned, “the efficiency of the device was characterized. It would be the capability of the device for generating monodispersed droplets.

Answer:

The efficiency of the droplet-forming microdevice was measured with the ability to generate monodispersity, which was demonstrated in Figure 3 AB, where the average size of 83 μm is observed.

6) (page 5, line 27) It was reported that the rate of single cell encapsulation in droplets is higher than 22% previously (Collins, D.J., Neild, A., DeMello, A., Liu, A.Q. and Ai, Y., 2015. The Poisson distribution and beyond: methods for microfluidic droplet production and single cell encapsulation. Lab on a Chip, 15(17), pp.3439-3459). Please clarify it.

Answer:

Our system has an efficiency like that of the works cited. A high efficiency (22%) of single cell encapsulation was achieved. To determine the optimal cell concentration number, the Poisson distribution law equation was used. In addition, the flow and density variables of the medium were added together with the transfection reagents.

7) (page5, line 38) After 24 hours of encapsulation, a high degree of cell death was observed; however, the authors proposed the microfluidic platform for long-term cell culture. Please explain further how the platform can be used for single cell long-term culture and transfection applications.

Answer:

It was considered that the system would not only be focused on gene editing systems such as PiggyBac transfection, viral systems or CRISPR-Cas9. By having versatility in micro-storage devices with levels of depth in manufacturing, it is possible to use hydrogels (Alginate, Matrigel or commercial) to maintain a single cell culture in the long term.

Our group is working on new research for single cell culture in hydrogels and gene editing using CRISPR-Cas9.

8) (page5, line 55) The authors showed the efficiency of transfection in the proposed platform. However, cell viability after transfection is one of the most important parameters for single cell transfection applications. Could the authors evaluate the cell viability after transfection in the proposed system?

Answer:

This recommendation was considered, and the cells were followed after transfection through 3 cell passages, demonstrating that the cells transfected by our method are still viable and can generate stable lines from them. . This is described on page 7 line 220, Results, segment 4.6 Post-transfection cell viability.

9) (Figure 4) It is not clear where droplets and cells are in figure 4A. A bright-field microscopy image on the area is necessary to be provided. Also, are the two images (green and red fluorescence) combined images? In Figure 4B, the authors showed the fluorescence intensity degradation to quantify the cell viability. However, the fluorescence intensity can be decreased by photobleaching. Hence, if the authors would evaluate the cell viability as a function of encapsulation time, the y-axis of the graph should be the number of cells (fluorescence dots).

Answer:

In figure 4, the cells were previously released from the droplets for their correct measurement. This was clarified in the manuscript. In Figure 4B, the fluorescence of each cell representing each point of the graph is taken. Given the reduction or absence of fluorescence, we determined that the cell lost viability or entered apoptosis. This was contrasted with the Texas network fluorescence channel, which determined whether the cell was indeed not viable.

10) English should be updated. The manuscript is hard to read.

English and grammar was checked and corrected in the manuscript.